# Effect of Farmland Scale on Farmers’ Application Behavior with Organic Fertilizer

**DOI:** 10.3390/ijerph19094967

**Published:** 2022-04-19

**Authors:** Yushi Chen, Xinhong Fu, Yuying Liu

**Affiliations:** 1College of Management, Sichuan Agricultural University, Chengdu 611130, China; 2020209051@stu.sicau.edu.cn; 2Sichuan Center for Rural Development Research, College of Management, Sichuan Agricultural University, Chengdu 611130, China

**Keywords:** organic fertilizer, farmland scale, instrument variable method, China, citrus farmer, heterogeneity

## Abstract

The application of organic fertilizer is an important measure for improving the quality of agricultural products and realizing the sustainable development of agriculture. The original small-scale family business model in China is no longer suitable for the development of modern agriculture. Large-scale agricultural business has become the mainstream trend, accompanied by the increasingly active land-transfer market. It is, therefore, necessary to determine whether farmland scale influences farmers’ organic fertilizer applications in China. Based on the analysis of the influence mechanism of farmland scale on farmers’ organic fertilizer application behaviors, empirical tests were conducted using survey data from 812 citrus farmers in Sichuan Province, China. The results show that the large-scale farmers are more likely to apply organic fertilizer and increase the application intensity than small-scale farmers and that the heterogeneity of farmers also affect their organic fertilizer application behavior. The results suggest that the government should encourage farmers to conduct moderate-scale management and promote their application behavior with organic fertilizer by improving awareness, strengthening education and training, and providing financial support.

## 1. Introduction

In the past few decades, chemical fertilizer has been thought of as the most direct and efficient way to improve productivity, which has resulted in the overuse of chemical fertilizer all over the world, especially in China. According to the FAO (Food and Agriculture Organization of the United Nations), China has been the country with the most chemical fertilizer use in the world for more than 30 consecutive years since 1984. In particular, Chinese farmers prefer to apply nitrogen fertilizer, phosphate fertilizer, and compound fertilizer [1]. However, excessive application of chemical fertilizer aggravates environmental pollution by harming soil, water, atmosphere, etc. [2,3]. The damage to the soil is manifested in soil acidification and compaction, the reduction of organic matter content, and heavy metal pollution [4,5,6]. In addition, the excessive application of chemical fertilizer leads to soil pollution that will further affect human health throughout the food chain [7]. For example, the excessive application of phosphate fertilizer leads to serious cadmium pollution, and the absorption of cadmium in food by the human body could cause bone disease, osteoporosis, and even cadmium poisoning [8]. Excessive application of nitrogen fertilizer leads to excessive accumulation of nitrite and nitrate in plants, and nitrates can be converted into nitrites after eating. Nitrites can combine with amines to induce various digestive system cancers, and can also reduce the oxygen-carrying capacity of the blood, thereby inducing methemoglobinemia and death by suffocation in severe cases [9]. In terms of pollution to water bodies, the extensive application of chemical fertilizers causes nitrogen, phosphorus, and other nutrients to enter bodies of water in large quantities, causing eutrophication of the water and nitrate pollution of the groundwater [10,11]. The impact of chemical fertilizers on the atmospheric environment is mainly due to nitrogen fertilizers. The volatilization of ammonia and the release of nitrogen oxide increase the nitrogen content in the atmosphere, which not only causes the greenhouse effect but also the destruction of the ozone layer [12]. Therefore, in order to prevent further deterioration of rural environments and to ensure the quality and safety of agricultural products, it is imperative to reduce the input of chemical fertilizer [13].

There are a variety of methods to achieve a reduction in chemical fertilizer application, among which are replacing chemical fertilizer with organic fertilizer, soil testing, formula fertilization, and carrying out a fallow rotation system [14,15]. Based on field experiments and farmer survey data, it is highly feasible to replace chemical fertilizers with organic fertilizers [16,17]; however, this replacement is not a complete replacement but a partial replacement in order to reduce excessive chemical fertilizer to an appropriate amount. According to the relevant Chinese standards, organic fertilizer is defined as carbonaceous materials mainly derived from plants and (or) animals and applied to soil to provide plant nutrition as the main function. The organic fertilizer mentioned here refers to commercial organic fertilizer that is not homemade that is purchased by farmers from other places. There is no denying that organic fertilizer contains a lot of organic matter and nutrition, which can significantly improve soil fertility, whereas chemical fertilizer only provides inorganic nutrients and its long-term application can lead to malnutrition. [18,19]. However, organic fertilizer also has its disadvantages. Although the unit prices of organic fertilizer and chemical fertilizer are not much different, organic fertilizer has a low nutrient content and requires a lot of fertilization, whereas chemical fertilizer has a high nutrient content, less dosage, and quicker effects [20]. Developed countries such as the United States and Japan began to advocate for the application of organic fertilizer in the production of food crops and to carry out organic agriculture at the beginning of this century and even in the 1980s and 1990s [21]. However, during these periods, China and other developing countries rarely used commercial organic fertilizer or even had relevant standards for commercial organic fertilizer [22]. Until 2017, the Ministry of Agriculture and Rural Affairs of China ran the “Five Major Actions for Agricultural Green Development” program, which firstly carried out demonstrations on how to replace chemical fertilizer with an organic fertilizer in areas that produce fruits, vegetables, and tea [23]. On the one hand, China chose to carry out the pilot project on fruit, vegetable, and tea due to the fact that the chemical fertilizer applied to these crops was significantly high; on the other hand, as fresh agricultural products, the quality and safety of fruit, vegetables, and tea have attracted more attention, and the quality-improvement effects and income-increase benefits have been more prominent since the substitution of organic fertilizer for chemical fertilizer. In the same year, a technical plan for replacing chemical fertilizer with organic fertilizer was introduced to effectively guide farmers in specific fertilization. Since 2017, in order to promote the development of the organic fertilizer industry, various provinces in China have issued policies to subsidize and reward organic fertilizer production enterprises, organic fertilizer raw material operators, and organic fertilizer applications. In addition, farmers can receive subsidies ranging from USD 25 to 75 per ton for buying organic fertilizer according to provincial policies. Despite these advantages, farmers with different endowments have different decision-making behaviors in the application of organic fertilizer. For instance, a considerable amount of literature has been published on the internal factors that influence farmers’ organic fertilizer applications, such as farmers’ personal and household circumstances, e.g., education level, family labor force, and annual household income [24,25]. Scholars have also explored some external driving factors in the application of organic fertilizer such as a code of ethics, social pressure, and an agricultural subsidy policy [26,27].

Since the input of organic fertilizer is an immovable protective investment attached to farmland, scholars also pay attention to farmland characteristics such as farmland scale, farmland fragmentation, and farmland confirmation [28,29]. When it comes to farmland, it is necessary to mention the reform of the rural household contract responsibility system, which began in the early 1980s, that allocated farmland in accordance with the number of family members and greatly stimulated farmers’ enthusiasm for production [30]. However, this division rule has also led to serious problems such as farmland fragmentation and low benefit [31,32]. According to data released by the World Bank, the per capita arable land area in China is less than 0.1 ha, which is only 40% of the average global level. Furthermore, with economic and social developments, especially the continuous off-farm transfer of the agricultural labor force, problems such as farmland abandonment, farmland fragmentation, and the insufficient fertility of farmland have become increasingly prominent [33]. Therefore, the Chinese government has exerted efforts to change the status quo and has emphasized the development of appropriately scaled agricultural operations of various forms in its annual No.1 central document since 2012. Land transfer is a crucial means for strengthening moderate-scale agricultural operations [34], and more and more farmers transfer land for scaled and specialized production and management [35,36]. According to the data from the National Bureau of Statistics of China, the average size of an orchard was 1.39 ha in 2018. The proportion of farmers whose farmland management scale was more than 6.67 ha increased from 0.20% in 2003 to 0.60% in 2018. Large-scale farmers will account for an increasing proportion of agricultural producers in the future, a fact that has attracted the attention of scholars. However, there are different views on the effect of farmland scale on the application of organic fertilizer by farmers. Based on different theories, some scholars identify with the conventional idea that large-scale farmers are more likely to apply organic fertilizer [23,37]. To be specific, large-scale farmers could have lower average costs and gain higher earnings than small-scale farmers from applying organic fertilizer. Large-scale farmers also have non-economic incentives such as their reputations and other positive externalities, which stimulate them to apply organic fertilizer [38,39]. However, others consider that farmland scale did not have a significant impact on the adoption of organic fertilizer technology [40,41]. Considering that farmers expand farmland scale by engaging in land transfers, some scholars have even studied the effect of farmland scale on farmers’ organic fertilizer applications from the perspective of land transfer. For instance, Bambio and Agha [42] think that large-scale farmers have a lower willingness to carry out green production on the farmland compared to small-scale farmers, who have not engaged in land transfers as a result of issues such as the stability of the management right, the transfer period, and the rent.

This study attempts to contribute to the literature threefold. First, the research conclusions reached by various scholars are somewhat different, so our research on the relationship between farmland scale and farmers’ applications of organic fertilizer will have a marginal supplementary effect. In particular, we establish a theoretical analysis framework from the perspective of the capital owned by the farmers that is lacking in other literature. Second, in the selection of samples, we select citrus farmers from China. As one of the most important economic crops in the world, citrus has the largest international trade volume of all fruit and is the third-largest agricultural trade in the world, but it is not the focus of academic research. In recent years, the scale of China’s citrus industry has continued to grow. Data from the National Bureau of Statistics of China shows that the Chinese citrus-planting area and output in 2019 were 2.62 million hectares and 45.85 million tons, respectively, both ranking first in the world. It is worth noting that the application of chemical fertilizer in economic crops far exceeds that in food crops, especially vegetables and fruits. So, it is necessary to promote organic fertilizer instead of chemical fertilizer in citrus planting. Third, we pay attention to the potential endogeneity between farmland scale and organic fertilizer adoption, which has not been mentioned in other literature. Moreover, we use instrument variable methods to weaken the endogeneity caused by the reciprocal causation and obtain more robust estimation results.

The remainder of this paper is arranged as follows. Section 2 presents the theoretical analysis, data sources, and econometric approach, as well as the data’s descriptive statistics, and it is followed by Section 3 which presents the empirical results. Section 4 discusses the empirical results. Section 5 presents the conclusion and policy implications.

## 2. Materials and Methods

### 2.1. The Influence of Farmland Scale on Farmers’ Organic Fertilizer Applications

According to the theory of behavioral economics, farmers, as rational economic men pursuing the maximization of personal interests, should first consider the minimum cost and the maximum profit when considering whether to choose organic fertilizer [43]. In particular, this paper builds a theoretical analysis framework to explore the influence of the farmland scale on farmers’ applications of organic fertilizer (Figure 1).

The difference in the scale of farmland will bring about differences in the social status and resource acquisition of farmers, and then affect farmers’ adoption of new technologies [44]. Generally speaking, large-scale farmers have better resource acquisition ability, especially in the acquisition of market information and policy information. First of all, as consumers in the agricultural materials market, large-scale farmers can acquire a higher-quality screening ability and price negotiation ability through centralized or large-scale purchasing. In other words, large-scale farmers can obtain relatively high-quality and low-cost organic fertilizer when compared to small-scale farmers. Therefore, the probability of large-scale farmers applying organic fertilizer is greater than that of small-scale farmers. Secondly, as producers in the agricultural products market, large-scale farmers are able to interact more effectively with markets, and they will turn to high-quality and environmental-friendly production modes to satisfy customers’ needs. The long-term and effective application of organic fertilizer can improve the quality of agricultural products, thereby meeting consumer demand for high-quality agricultural products [45]. As a result, large-scale farmers are more likely to apply organic fertilizer in this regard. Besides, large-scale farmers have more opportunities to obtain technical services and policy support provided by the government, and can accept new ideas such as environmentally friendly and green production, and can then take action to master these new technologies and apply organic fertilizer more effectively [46]. However, small-scale farmers’ access to information is often restricted, which also hinders their application of organic fertilizer [47]. Based on the above analysis, we propose two hypotheses:

**Hypothesis** **1** **(H1).***Large-scale farmers are more likely to apply organic fertilizer than small-scale farmers*.

### 2.2. Heterogeneity Analysis on the Effect of Farmland Scale on Farmers’ Organic Fertilizer Applications

Previous studies have found that resource endowment is an important factor affecting farmers’ decision-making behaviors in the application of organic fertilizer [48]. There are significant differences in the impact of human capital, such as age, education level, and physical condition, on farmers’ willingness to apply organic fertilizer [49]. Compared to farmers with low education skills, farmers with higher education skills are more willing to apply organic fertilizer [50]. Social capital also results in significant differences in farmers’ application of organic fertilizer. Farmers with high social capital, such as village heads, find it easier to access information and resources than ordinary farmers, causing them to have a better understanding of the benefits of organic fertilizer [38]. In addition, there are differences in the application of organic fertilizers by farmers with different financial capital. Organic fertilizer application is a labor- and capital-intensive technology, where farmers need to spend a lot of money to purchase organic fertilizer and hire complementary labor to ensure the effective application of the fertilizer. The farmers who have difficulty in obtaining funds may not be able to support the related costs, leading them to be less likely to apply organic fertilizer.

The effect of farmland scale on farmers’ organic fertilizer applications is heterogeneous. After the scale of the farmland has been moderately expanded, farmers can access more resources for of agricultural production, including richer social networks, more learning opportunities, and the ability to discover the benefits of organic fertilizer and apply them in time. Therefore, compared to farmers with higher capital endowments, such as village heads and party secretaries, farmers with lower capital endowments have relatively higher social and economic benefits from expanding the scale of their farmland and are able to promote their application of organic fertilizer. When it comes to financial capital, however, it is different. The expansion of farmland brings not only an increase in the means of production, but also high farmland rents, which results in farmers with a strong financial ability being more likely to apply organic fertilizer [48]. Therefore, this paper proposes another hypothesis:

**Hypothesis** **2** **(H2).***There is heterogeneity of social capital, human capital, and financial capital in the effect of farmland scale on farmers’ organic fertilizer applications*.

### 2.3. Data Source

Citrus is the fruit with the largest planting area, the highest output, and the largest consumption in China. Sichuan province is located in the superior citrus region in the upper and middle reaches of the Yangtze River. According to data released by China’s agricultural sector, Sichuan’s citrus-planting area was over 400 thousand ha by the end of 2019, ranking second in China, and Sichuan has built the largest late-ripening citrus-production area in inland China. At present, Sichuan citrus is mainly distributed in the Chengdu Plain and East Sichuan, in Chengdu, Guang’an, Ziyang, Meishan, Luzhou, etc. Meishan City is the city with the largest citrus-planting area in Sichuan Province, with nearly 700,000 ha. Furthermore, eight counties in Sichuan were listed as national demonstration counties for the replacement of chemical fertilizer with organic fertilizer for fruit, vegetable, and tea crops in 2017, ranking first in the country. Among them, Pujiang County, Xichong County, Guang’an District of Guang’an City, Anyue County, and Danling County are the demonstration counties for citrus organic fertilizer to replace chemical fertilizer.

The data used in this paper come from a survey of citrus farmers conducted between July and August 2020 in Sichuan province, China. Important sampling and stratified random sampling were used to select the sample households for the empirical analysis. In the first stage, the five cities of Chengdu, Ziyang, Meishan, Nanchong, Neijiang, and Yibin were selected since they are representative of citrus production at the provincial level. The total citrus production in the sample area accounted for more than 60% of the total production in Sichuan Province. Second, 1 to 3 counties in each city were selected based on the intensity of citrus production in those selected cities. Third, 4 to 5 villages from each sample county were randomly selected. Finally, 20 to 25 citrus farmers in each village were randomly selected. The survey was conducted using face-to-face questionnaire interviews with citrus farmers. The content of the questionnaire included the basic characteristics of household heads and their families, agricultural production and management characteristics, cognitive characteristics, social participation, and so on. The agricultural production characteristics included the scale of farmland and the application of organic fertilizer to be studied in this paper. According to its processing and nutrient status, commercial organic fertilizer can be divided into refined organic fertilizer, organic compound fertilizer, and bio-organic fertilizer. The organic fertilizer mentioned in the survey refers to refined organic fertilizers, including commodities with a total nutrient content of ≥4% and organic matter of ≥30% purchased by farmers from agricultural material distribution stores, e-commerce platforms, farms, agricultural service stations, etc. The final dataset comprises a total sample of 812 citrus farmers.

### 2.4. Methodology

#### 2.4.1. Variable Selection

The dependent variables are selected to represent farmers’ organic fertilizer application behavior in the empirical model. The first is “whether farmers apply organic fertilizer”, whereas the second is “the intensity of applying organic fertilizer”. The measurement of “whether farmers apply organic fertilizer” uses the binary valuation method. If the farmers apply organic fertilizer, the value is 1. On the other hand, farmers do not apply organic fertilizer if the value is 0. In addition, “the intensity of applying organic fertilizer” is expressed in terms of the ratio of organic fertilizer to total fertilizer application. In Table 1, nearly 70% of farmers chose to apply organic fertilizer in our sample. The average intensity of organic fertilizer application in the sample is 24.5%, indicating that a majority of the farmers are still in a trial stage, rather than using large-scale application of organic fertilizer.

This paper selects farmland scale as the independent variable. The average citrus planting scale in the sample is 1.45 ha, but most of the sample households in our survey still plant on a small scale of less than 0.67 ha. The larger average planting scale is mostly related to regional heterogeneities such as local terrain and land transfer. The majority of citrus plantations in our survey are located on mountains or hills where the planting scale is larger than that in plain areas. In addition, large-scale citrus farmers transferred in the farmland of several farmers for their own management, which also resulted in a larger average farmland scale in the sample.

Referring to the relevant literature [51,52,53], we chose the farmer’s personal and household characteristics, location characteristics, cognition of organic fertilizer application, and external participation as the control variables. Farmer’s personal and household characteristic includes education, planting year, and agricultural labor. The average education level and planting year is 7.15 and 13.79, respectively. This shows that most farmers have lower education skills and have been engaged in agricultural production for a long time. In addition, there are few family members engaged in agricultural production, usually the elderly. Location characteristics involve terrain and distance to market. The terrain of citrus-planting areas is dominated by hills and mountains, which also leads to a long distance between home and the market, with an average of 3.51 km. Cognition of organic fertilizer application includes environmental awareness and economic cognition, and external participation includes joining farmers’ cooperatives and training time. From the variables’ mean, the level of farmers’ environmental awareness of organic fertilizer is higher than that of their economic cognition. Nearly two-thirds of farmers have participated in farmers’ cooperatives, and the average number of training sessions is 2.5.

Table 2 presents the means of selected variables by organic fertilizer application category (1 = farmer has applied organic fertilizer and 0 = otherwise) and group mean differences. It shows that organic fertilizer applicators have larger farmland scales than non-applicators. Organic fertilizer applicators are better-educated and have more planting experience compared with non-applicators. There is, however, no significant difference in agricultural labor and terrain between organic fertilizer applicators and non-applicators. The mean comparisons indicate that organic fertilizer applicators and non-applicators are significantly different with respect to distance, environmental awareness, and economic cognition. Table 2 also shows that for organic fertilizer applicators, the percentage joining farmers’ cooperatives and the number participating in organic fertilizer-related training is significantly higher than for those of non-applicators. However, further empirical tests are needed to quantitatively analyze the effects of farmland scale and different control variables on organic fertilizer application behaviors of citrus farmers.

#### 2.4.2. Model Specification

In our study, two dependent variables are selected to represent farmers’ organic fertilizer application behavior in the empirical model. The first is whether farmers apply organic fertilizer, whereas the second is the intensity of applying organic fertilizer. Since whether farmers apply organic fertilizer is a binary variable, a Probit model was selected as a benchmark for estimation. The intensity of applying organic fertilizer is the ratio of organic fertilizer to total fertilizer application. Since the value of the intensity of applying organic fertilizer is between 0 and 1, a Tobit model was selected as a benchmark for estimation.

Before performing the regression, a multicollinearity test should be conducted to avoid strong correlativity between the independent variables. The VIF values of all independent variables are less than 5 and the average VIF is 1.16, which denotes that there exists no multicollinearity among the variables. After passing the multicollinearity test, binary Probit regression and Tobit regression are applied to the sample data. The forms of the two models are as follows:$$Logit\left(p\right)=Ln\left(\frac{p}{1-p}\right)=\beta_{0}+\beta_{s}scale_{i}+\beta x_{i}+\varepsilon_{1i}$$
$$The\;intensity_{i}=\gamma_{0}+\gamma_{s}scale_{i}+\gamma x_{i}+\varepsilon_{2i}$$
where p represents the probability that farmers choose to apply organic fertilizer; $The\;intensity_{i}$ represents the organic fertilizer intensity of total fertilizer application; $scale_{i}$ indicates citrus-planting scale; and $x_{i}$ represents the other control variables. Besides, β and γ are the regression coefficients that are estimated, and $\varepsilon_{i}$ is the random disturbance term.

However, there is reciprocal causation between farmland scale and farmers’ organic fertilizer application behavior. On the one hand, the expansion of farmland scale improves the economies of scale and can effectively impel citrus farmers to apply organic fertilizer for long-term productive investment; on the other hand, the application of organic fertilizer can improve soil fertility and thus increase the quality and yield of crops, encouraging farmers to expand farmland scale to obtain more benefits. A two-stage instrumental variable model (IV-Probit and IV-Tobit) can solve the endogeneity in the benchmark model caused by reciprocal causation. The variable “the development of local citrus industry” is selected as the instrumental variable for the endogenous variable “farmland scale”. If the local citrus industry is developing well, farmers will be more likely to transfer land with better profits, and then their planting scale will become larger. Therefore, there is a positive relationship between the development of the local land transaction market and farmland scale. Moreover, there is no research showing that the development of the local land transaction market has an impact on the application of organic fertilizer by farmers.

In order to further analyze the effect of farmers’ heterogeneity, we select indicator variables able to obtain market price, technology, policy and other information, years of education, and ability to obtain funds from financial institutions to represent farmers’ social capital, human capital, and financial capital. The size of these three variables is reflected in the Likert five-level scale, which gradually increases from 1 to 5. According to differences in social capital, that is, the size of the corresponding indicators of social capital, the whole sample is divided into two sub-samples with the average value as the limit. Human capital and financial capital are also grouped in the same way. Then, group regression models are performed.

## 3. Results and Analysis

### 3.1. The Impact of Farmland Scale on Citrus Farmers’ Organic Fertilizer Applications

All the models passed the chi-square test (LR Chi^2^) at the 1% significance level, which indicates that the models fit well. The Pseudo R^2^ of Probit (II) increased by 0.9% compared with Probit (I) after farmland scale was added. The Pseudo R^2^ of Tobit (II) increased by 0.1% compared with Tobit (I) after farmland scale was added. It is evident that the explanatory power of the benchmark model is enhanced by adding the main dependent variables. This indicates that farmland scale is the key factor influencing the application of organic fertilizer by citrus farmers. The estimated results are listed in Table 3.

The coefficient of farmland scale in Probit (II) is 0.005, which is statistically significant at the 5% level. The coefficient of farmland scale in Tobit (II) is 0.023 at the 5% significance level. The results verify hypothesis H1, indicating that the possibility and the intensity of large-scale farmers applying organic fertilizer are both greater than those of small-scale farmers. In other words, if farmers expand the scale of their farmland, the possibility and intensity of organic fertilizer application will increase accordingly. Among the control variables, education level and environmental awareness both have significantly positive effects on the probability and intensity of a farmer applying organic fertilizer, and they all passed the 1% and 5% significance level tests. Distance to market has a negative effect on the probability and intensity of citrus farmers’ application of organic fertilizer at the significance level of 1%. This shows that farmers close to the market are more likely to apply organic fertilizer and increase the application intensity. Farmers’ cooperatives and training have significant positive effects on the probability and intensity of a farmer applying organic fertilizer. This indicates that farmers who join cooperatives or participate in training more often have a better chance to learn about organic fertilizer and its application technology, causing them to then apply the organic fertilizer.

### 3.2. Estimated Results of the IV-Probit Model and the IV-Tobit Model

Further testing the rationality of the instrumental variable, the results in Table 4 show that the impact of the instrumental variable on the endogenous variable farmland scale passed the significance test at the 1% level, with strong explanatory power. The Cragg–Donald Wald F value is 12.35, which is greater than 10, indicating that there is no problem with weak instrumental variables. The IV-Probit model and IV-Tobit model both reject the null hypothesis at the significance level of 1%, indicating that there are exogenous variables in the models. The farmland scale variable still has an impact on citrus farmers’ organic fertilizer application behaviors in the IV-Probit model and the IV-Tobit model, and the direction is completely consistent with the benchmark model. It can be concluded that farmland scale still has a positive effect on the probability and intensity of farmers’ organic fertilizer applications when the endogeneity issues have been eliminated.

### 3.3. Robustness Check

To further check the effect of farmland scale on organic fertilizer application, we randomly selected 85% of the samples to form a sub-sample with a sample size of 690 and performed quadratic regression to test the robustness of the estimation results. It can be seen in Table 5 that the significance and influence of farmland scale in the sub-sample model estimation are consistent with those in the benchmark models, indicating that farmland scale has a positive effect on the probability and intensity of farmers’ organic fertilizer applications. The above results show that the main empirical results of this paper are relatively robust.

## 4. Discussion

Although organic fertilizer application is promoted as a preferred approach for improving the quality of agricultural products and reducing the application of chemical fertilizer, its application rate remains low in China. There is still a need to promote the adoption of organic fertilizer. Although recent studies have shown that farmland scale has an impact on the material input of farmers, the positive or negative impacts have not been agreed to, especially in the input of fertilizer. In this paper, we have put forward our original ideas by combining theory with empirical analysis. Through empirical testing, our findings are consistent with Lu et al. [48], who found that the unit value of agricultural products produced by large-scale farmers is higher compared with that produced by small-scale farmers, and in turn, this increases large-scale farmers’ long-term investment in agriculture, including organic fertilizer application. The higher unit value of agricultural products is reflected in the reduction of the average cost of large-scale procurement by large-scale farmers, and the expected higher price of green production and catering to the green food market [54]. In addition to the lower costs and higher benefits brought by large-scale production, large-scale farmers tend to have more opportunities for acquiring technical services provided by the government and other social organizations, which could encourage the adoption of organic fertilizer by increasing the potential benefits of organic fertilizer application [55]. Farmers with larger farmland scale participated in training far more times than small-scale farmers, which is confirmed in our survey. Even a sample of large-scale citrus farmers participated in training as many as 50 times in 2019. However, participation in the training is only part of the reason. Large-scale farmers have sufficient learning and financial ability for subsequent organic fertilizer application [56], so their organic fertilizer application intensity is higher than that of small-scale farmers. Among the control variables, education level and planting year reflect the farmers’ cognitive and technical learning abilities with regard to organic fertilizer to a certain extent. The stronger the cognitive and learning ability of the farmers, the more likely they are to apply organic fertilizer. Farmers’ cooperatives and training have both had significant positive effects on citrus farmers’ organic fertilizer application behaviors, which suggests that the more farmers understand market demand and the information and technologies connected with organic fertilizer, the more likely they will be to apply organic fertilizer [57,58]. However, the regression result of the distance to market is in contrast to that of Mponela and his colleagues [59]. They pointed out that farmers can get higher returns from selling their products in formal markets far away from them, leading to their application of organic fertilizer. From our perspective, the higher return may not be enough to offset the higher costs of long-distance transportation, especially in hilly and mountainous areas [60].

In order to further analyze the effect of farmers’ heterogeneity, we perform group regressions on the basis of the farmers’ social capital, human capital, and financial capital. The regression results are shown in Table 6. The coefficient of farmland scale in Probit (V) and Tobit (V) are 0.008 and 0.078, respectively, which are both statistically significant at the 5% level. The results show that for farmers with lower social capital, the expansion of farmland scale can impel them to apply organic fertilizer and increase the application intensity of organic fertilizer. The possible explanation is that with the expansion of farmland scale, farmers with low social capital have wider information channels, which will deepen their cognition of green production, strengthen their bargaining power, and thus encourage them to apply organic fertilizer. However, for farmers with higher social capital, the expansion of farmland scale may not be the main method to promote the application of organic fertilizer. The coefficient of farmland scale in Probit (VI) and Probit (VII) are 0.005 and 0.007, respectively, at the 10% significance level, indicating that large-scale farmers are more likely to apply organic fertilizer regardless of their human capital. In addition, the expansion of farmland scale can motivate farmers with higher human capital to increase the application intensity of organic fertilizer, but it has no significant impact on the intensity of organic fertilizer application by farmers with lower human capital. Farmers with higher human capital have a better understanding of organic fertilizer [61], and are aware that if the intensity of organic fertilizer is not at a certain level, its beneficial effects will not be significant. The coefficient of farmland scale in Probit (VIII) and Tobit (VIII) are 0.005 and 0.019, respectively, which are statistically significant at the 10% level. The results show that for farmers with higher financial capital, the expansion of farmland scale can motivate them to apply organic fertilizer and increase the application intensity of organic fertilizer. The possible explanation is that these farmers can support a variety of expenses after expanding their farmland scale, such as agricultural materials and farmland rent brought in by land transfers. Nevertheless, for farmers with lower financial capital, the expansion of farmland scale cannot significantly promote their application of organic fertilizer, possibly due to uncertainty about their future earnings, which makes them reluctant to add new costs and risks by applying organic fertilizer [62,63]. To sum up, the above results show that the impact of farmland scale on farmers’ organic fertilizer applications is indeed heterogeneous in terms of social capital, human capital, and social capital, which confirms Hypothesis 2.

## 5. Conclusions

In this paper, we theoretically and empirically analyzed the impact of farmland scale on citrus farmers’ organic fertilizer application behaviors. We employed the Probit and Tobit models to analyze the probability and intensity of organic fertilizer application with field survey data from 813 citrus growers in the citrus-producing areas of Sichuan Province in the summer of 2020, and we employed instrument variable methods to solve the endogeneity problem of mutual causation. To further understand whether farmers’ heterogeneity plays a role in the impact of farmland scale on farmers’ organic fertilizer applications, we conducted group regression according to the level of the farmers’ social, human, and financial capital. Through the analysis of theoretical and empirical results, we came to the following conclusions. First, evidence shows that farmland scale has a significant positive effect on the probability and intensity of citrus farmers’ organic fertilizer applications. In other words, large-scale farmers are more likely to apply organic fertilizer and even enhance the application intensity. Second, the heterogeneity of farmers’ social, human, and financial capital has an impact on farmers’ organic fertilizer applications. Due to the increase in social capital that scale brings, large-scale farmers have easier access to organic fertilizer information and techniques to facilitate their application. The expansion of farmland scale could prompt farmers with higher human capital to increase the intensity of organic fertilizer, rather than those with lower human capital. For farmers with higher financial capital, the probability and intensity of applying organic fertilizer is higher than those with lower financial capital because they have more funds to support the various expenses related to organic fertilizer application.

Since the application of organic fertilizer in agricultural production is conducive to the protection of soil fertility and the sustainable development of agriculture, our results offer important policy implications concerning the interactions between farmland scale and organic fertilizer application. First, considering that large-scale farmers are more likely to apply organic fertilizer, the government should properly promote land transfer and the continuous-scale operation of farmland. However, it is essential to avoid blindly expanding the scale of operation. After all, large-scale farmland rent cannot be ignored. In addition, the guiding and leadership roles played by large-scale farmers should be utilized to encourage small-scale farmers to carry out environmentally friendly production such as soil experiment formula fertilization, green fertilizer planting, organic fertilizer application, soil improvement, etc. Second, in view of farmers’ heterogeneity in the effect of farmland scale on citrus farmers’ organic fertilizer applications, the government should focus on enhancing farmers’ awareness and strengthening farmers’ abilities. It is necessary for the government to strengthen the publicity, promotion, and demonstration of organic fertilizer in order to increase farmers’ enthusiasm for applying organic fertilizer. Furthermore, the government should expand education and training for farmers and encourage them to join farmers’ cooperatives to facilitate communication and learning. Only when farmers master the technology related to organic fertilizer application will they take action to implement it. Additionally, the government should give farmers more financial support, such as relaxing loan standards and increasing subsidies for farmers for green production.

However, limitations inevitably exist in this study. Firstly, the study focuses only on specific types of farmers within a single region. Studies focusing on other crops and other regions are necessary to obtain a better understanding of the heterogeneous impacts of farmland scale on farmers’ organic fertilizer application behaviors in a broad context. Due to data availability, some potential explanatory variables are missing, such as total household income, soil condition, specific subsidy amount, etc. It is important to note that the research also omits economic factors (such as the price of organic fertilizer, the price of chemical fertilizer, etc.). It is possible that manpower and material costs related to organic fertilizer application are more important to farmers than its actual unit price; however, there is no relevant data available. Hence, future studies could incorporate these factors and provide a holistic understanding of the relationship between farmland scale and farmers’ organic fertilizer application behaviors. It also needs to be noted that all data rely on recall and statements from the interviewed farmers and are therefore susceptible to biases such as recall bias and social desirability bias.

## Figures and Tables

**Figure 1 ijerph-19-04967-f001:**
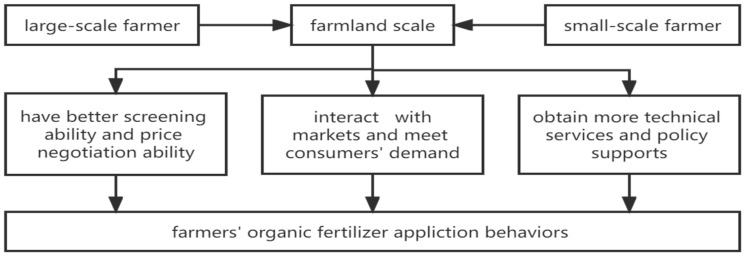
Analytical framework.

**Table 1 ijerph-19-04967-t001:** Variable definition and statistical descriptive analysis.

Variable Name	Variable Definition	Mean	S.D.
Dependent Variables			
Organic fertilizer application	1 = farmer has applied organic fertilizer; 0 = no	0.70	0.46
The intensity of application	The ratio of organic fertilizer in total fertilizer application (%)	24.55	25.09
**Independent variable**			
Farmland scale	Total citrus planting area (ha)	1.45	5.25
**Control variables**			
Education level	Farmer’s years of education (years)	7.15	3.62
Planting year	Farmer’s years of citrus planting (years)	13.79	10.69
Agricultural labor	The number of household agricultural laborers	2.05	1.00
Terrain	1 = plain; 2 = hill; 3 = mountain	2.01	0.43
Distance to market	Distance from farmer’s home to market (km)	3.51	2.91
Environmental awareness	The application of organic fertilizer can improve the environment (1 = strongly disagree; 2 = comparatively disagree; 3 = general; 4 = agree; 5 = strongly agree)	4.02	0.69
Economic cognition	The application of organic fertilizer makes citrus a good price (1 = strongly disagree; 2 = comparatively disagree; 3 = general; 4 = agree; 5 = strongly agree)	3.78	0.82
Farmers’ cooperative	1 = farmer has joined farmers’ cooperative; 0 = no	0.66	0.48
Training	The number of organic fertilizer related training sessions (times)	2.50	3.38

**Table 2 ijerph-19-04967-t002:** Mean difference of the variables used in the sample selection equation.

Variable	Organic Fertilizer Applicator	Non-Applicator	Difference
Farmland scale	28.243 (3.918)	7.016 (1.110)	21.227 ***
Education level	7.758 (0.143)	5.761 (0.239)	1.997 ***
Planting year	14.776 (0.453)	11.518 (0.650)	3.258 ***
Agricultural labor	2.064 (0.045)	2.024 (0.051)	0.039
Terrain	1.996 (0.019)	2.045 (0.024)	−0.048
Distance to market	3.228 (0.116)	4.159 (0.199)	−0.932 ***
Environmental awareness	4.117 (0.028)	3.810 (0.044)	0.307 ***
Economic cognition	3.878 (0.034)	3.571 (0.052)	0.307 ***
Farmers’ cooperative	0.708 (0.019)	0.538 (0.032)	0.170 ***
Training	3.046 (0.160)	1.239 (0.095)	1.807 ***
Observations	565	247	

Note: ***, ** and * indicate statistical significance at the 1%, 5% and 10% levels, respectively.

**Table 3 ijerph-19-04967-t003:** Estimation results of the impact of farmland scale on citrus farmers’ organic fertilizer applications.

Variable	Probit (I)	Probit (II)	Tobit (I)	Tobit (II)
Farmland scale		0.005 ** (0.002)		0.023 ** (0.011)
Education level	0.043 *** (0.016)	0.034 ** (0.016)	0.788 *** (0.248)	0.720 *** (0.249)
Planting year	0.014 *** (0.005)	0.015 *** (0.005)	0.105 (0.080)	0.120 (0.081)
Agricultural labor	−0.036 (0.055)	−0.070 (0.061)	−0.045 (0.823)	−0.455 (0.843)
Terrain	0.079 (0.124)	0.070 (0.126)	7.019 *** (1.998)	6.786 *** (1.996)
Distance to market	−0.072 *** (0.017)	−0.076 *** (0.018)	−0.888 *** (0.286)	−0.872 *** (0.286)
Environmental awareness	0.221 *** (0.082)	0.196 ** (0.083)	3.736 *** (1.350)	3.459 ** (1.353)
Economic cognition	0.070 (0.066)	0.077 (0.067)	−2.328 ** (1.107)	−2.277 ** (1.104)
Farmers’ cooperative	0.251 ** (0.109)	0.235 ** (0.110)	7.948 *** (1.800)	7.766 *** (1.797)
Training	0.188 *** (0.030)	0.182 *** (0.031)	1.596 *** (0.254)	1.553 *** (0.254)
Observations	812	812	812	812
LR chi^2^	159.88 ***	168.13 ***	127.98 ***	132.39 ***
Pseudo R^2^	0.160	0.169	0.017	0.018

Note: ***, ** and * indicate statistical significance at the 1%, 5%, and 10% levels, respectively.

**Table 4 ijerph-19-04967-t004:** Estimation results of IV-Probit and IV-Tobit.

Variable	IV-Probit	IV-Tobit
Farmland scale	0.015 *** (0.001)	0.473 *** (0.180)
Education level	−0.014 (0.0155)	−0.088 (0.729)
Planting year	0.015 *** (0.004)	0.505 ** (0.196)
Agricultural labor	−0.237 *** (0.047)	−8.723 ** (3.603)
Terrain	−0.072 (0.103)	3.800 (4.354)
Distance to market	−0.034 * (0.019)	−1.115 * (0.589)
Environmental awareness	−0.026 (0.080)	−0.188 (3.455)
Economic cognition	0.065 (0.055)	−0.687 (2.239)
Farmers’ cooperative	0.044 (0.100)	7.618 ** (3.863)
Training	0.077 ** (0.035)	1.182 ** (0.595)
Observations	812	812
Wald chi^2^	195.07 ***	78.63 ***
Exogeneity test (Chi^2^)	13.97 ***	10.11 ***

Note: ***, ** and * indicate statistical significance at the 1%, 5%, and 10% levels, respectively.

**Table 5 ijerph-19-04967-t005:** The results of the robustness check.

Variable	Probit (III)	Tobit (III)
Farmland scale	0.004 ** (0.002)	0.021 * (0.012)
Education level	0.027 (0.017)	0.612 ** (0.265)
Planting year	0.012 ** (0.005)	0.111 (0.088)
Agricultural labor	−0.074 (0.068)	−0.685 (0.955)
Terrain	−0.045 (0.136)	5.934 *** (2.169)
Distance to market	−0.077 *** (0.019)	−0.922 *** (0.312)
Environmental awareness	0.266 ** (0.090)	4.963 *** (1.475)
Economic cognition	0.088 (0.072)	−2.978 ** (1.210)
Farmers’ cooperative	0.193 (0.118)	7.231 *** (1.934)
Training	0.197 *** (0.034)	1.511 *** (0.263)
Observation	690	690
LR chi^2^	151.11 ***	113.40 ***
Pseudo R^2^	0.174	0.018

Note: ***, ** and * indicate statistical significance at the 1%, 5%, and 10% levels, respectively.

**Table 6 ijerph-19-04967-t006:** Estimation results of group regression model on different dimensions.

Social Capital	Higher Level		Lower Level	
	Probit (IV)	Tobit (IV)	Probit (V)	Tobit (V)
Farmland scale	0.001	0.007	0.008 **	0.078 **
	(0.002)	(0.012)	(0.004)	(0.030)
Control variables	Yes	Yes	Yes	Yes
Observation	413	413	399	399
LR chi^2^	49.21 ***	56.14 ***	72.53 ***	48.37 ***
Pseudo R^2^	0.144	0.015	0.132	0.014
**Human Capital**	**Higher Level**		**Lower Level**	
	**Probit (VI)**	**Tobit (VI)**	**Probit (VII)**	**Tobit (VII)**
Farmland scale	0.005 *	0.024 *	0.007 *	0.026
	(0.003)	(0.013)	(0.004)	(0.020)
Control variables	Yes	Yes	Yes	Yes
Observation	382	382	430	430
LR chi^2^	52.99 ***	75.73 ***	84.01 ***	46.35 ***
Pseudo R^2^	0.146	0.022	0.144	0.012
**Financial Capital**	**Higher Level**		**Lower Level**	
	**Probit (VIII)**	**Tobit (VIII)**	**Probit (IX)**	**Tobit (IX)**
Farmland scale	0.005 **	0.019 *	0.003	−0.013
	(0.003)	(0.011)	(0.006)	(0.087)
Control variables	Yes	Yes	Yes	Yes
Observation	368	368	444	444
LR chi^2^	64.19 ***	88.55 ***	88.38 ***	52.09 ***
Pseudo R^2^	0.174	0.026	0.149	0.013

Note: ***, ** and * indicate statistical significance at the 1%, 5%, and 10% levels, respectively.

## Data Availability

The data that support the findings of this study are available from the corresponding author upon reasonable request.

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
