# Peer review of "Effect of Farmland Scale on Farmers’ Application Behavior with Organic Fertilizer"

_ijerph, 2022, doi:10.3390/ijerph19094967_

Round 1

Reviewer 1 Report

The paper entitled "Effect of Farmland Scale on Farmers’ Application Behavior of Organic Fertilizer" by Yushi Chen et al is an interesting study and I think that could be considered for publication after minor to moderate revision.

I recommend that authors should review the Introduction section and include some aspects from literature regarding the negative effect of intensive chemical fertilization. The authors made some assessments regarding this topic referring just to heavy metal accumulation but the negative influence of chemical fertilization relies primarily in  issues associated with soil, water and air pollution. Nonetheless that there are no mentions regarding which are the most used chemical fertilizers and therefore which are the most problematic nutrients in terms of their huge accumulation in soils and overall natural environment and human health. I would recommend also to define more precisely the organic fertilizers and make some assessments regarding the most used organic fertilizers in China along with the presentation of their most important benefits such that the reader can understand the difference between the two types of fertilization and why it is so important to expand the use of organic fertilization detrimental to the use of chemical fertilization.

It is not clear for me if some parts from Section 2. Materials and Methods are in the right place. For instance the information found in 2.4. Methodology/2.4.1. Variable Selection looks more like results for me and thus I am not sure if these data would not fit better in Section 3. Results?
Also the first phrase from Section 3. Results and analysis I think would suit better in the previous section related to Material and method?

I recommend that authors pay attention to minor spell checks, for example on Page 5 – “It located”– it is located; Page 11, Line 21  - “the above results show” – did you mean the bellow results, from Table 6? The same confusion is also found in page 9, before table 5 – “The above results show that the main empirical results of this paper are relatively robust.” – I think that you mean the bellow results, meaning from table 5?

I would introduce Section 6 in Discussions section or even in Results.  I don’t think that suits to be introduced as an individual section after the Conclusion section, which should be the last section of the manuscript (presenting the main findings of the research).

Reviewer 2 Report

Reviewer report for paper entitled "Effect of Farmland Scale on Farmers’ Application Behavior of Organic Fertilizer". The study was was designed to understand in depth whether the plantation size can influence the decision of using organic fertilizer or not. Study was carried out from survey data from 812 citrus plantation in Sichuan province in China. The study is well designed but needs some improvement as follows: 

  • provide the full name of the abbreviation (FAOSTAT) when used first time in text.
  • Need to provide reference for the following sentence "In China, the per capita arable land area is less than 0.1 ha, which is only 40% of the world's average level". 
  • In introduction part, need to provide more information about the preferred use of organic fertilizer in previous studies for which plantation? and countries? 
  • The mean different variables used in this study is well designed (Table 2). However, for this type of study, its important to provide more information about sample selection and questionnaire, pilot study of any and more information about research instrument and the method of results validation. 
  • The results are well presented and the well discussed. However, authors need to highlight the limitation of this research and how this can be applied for other plantation. 
  • Conclusions need to be improved to highlight the limitations of this work as well. 

Reviewer 3 Report

The reviewed article concerns an important research problem. The methods used are appropriate/ suitable. The introduction and justification of the research problem are  appropriate.

I have one major reservation

Unfortunately, the authors omitted one of the most important factors, i.e. fertilizer prices and their "efficiency" (i.e. the impact on yields). If conventional fertilizers are cheaper and as effective as organic fertilizers, small farmers will choose conventional fertilizers. Only the "bigger" ones can afford "pro-ecological" production. Without this variable, the studies are incomplete (and reliable ??).

If there are large differences in fertilizer prices, the interpretation of the results should be different. Please explain why this aspect has not been tested. Please complete the information regarding the comparison of prices of conventional and organic fertilizers.

What organic fertilizers do the authors mean?

The authors omit quite an important issue, i.e. the availability of an organic fertilizer for large farms. Is it "easy" to buy large amounts of organic fertilizer in China?

Methodical aspects

The research was based on random selection. Did the Authors check whether the size of the research sample is sufficiently large to make the research representative? What statistical tests were used to determine the "Mean difference of the variables used in the sample selection equation" (Table 2). The hypotheses were formulated in such a way that, in principle, they could be verified with statistical tests.

Other comments and suggestions

In the introduction, the authors provide information about  the area of agricultural land per capita. However, it may be better to present information about the average farm’s size.

I do not entirely agree with this claim: “Based on economies of scale theory and prospect theory respectively, some scholars identify with the conventional idea that large-scale farmers are more likely to apply organic fertilizer”.

Should it be related to the "economies of scale theory"? Lower unit costs (thanks to the large scale) will not make the farmer more willing to use organic fertilizers. According to the theory of economics (especially neoclassical), the farmer will use such fertilizers and in sizes that will give the highest profit / income.

Please indicate the limitations related to the obtained results.

Please provide more information about the study area.

Round 2

Reviewer 3 Report

Thanks to the authors for their very comprehensive answers. I am glad that the suggestions were taken seriously.

I still believe that the price, and in fact, the monetary effect of using fertilizers, will be decisive. Organic fertilizers work differently than mineral fertilizers, but it is possible to calculate how much 1 kg of NPK costs in organic and mineral fertilizers. I am convinced that Chinese farmers will also calculate which fertilizer contributes to higher profits. I do not deny the legitimacy of promoting organic fertilizers, but in the realities of a market economy, it is economic efficiency that matters the most.

Of course, organic fertilizers work differently than mineral fertilizers ... .. Nevertheless, mineral fertilizers, if properly applied (appropriate dates, appropriate doses of individual components: N, P, K, Mg, etc.) will not have a negative impact on the quality of the crop. A separate issue is the assessment of how they affect human health and the natural environment ... but it is not the subject of the authors' considerations.

To sum up, since the authors do not want to include in the analyzes the issue of the efficiency of a fertilizer application (fertilizer prices), I would like to ask you to clearly emphasize that the manuscript omits economic factors (differences in the efficiency of fertilizer application).

This is a very important information and is also seen as an important limitation of the manuscript.
